# Impact of Nirsevimab Immunization on Pediatric Hospitalization Rates: A Systematic Review and Meta-Analysis (2024)

**DOI:** 10.3390/vaccines12060640

**Published:** 2024-06-08

**Authors:** Matteo Riccò, Antonio Cascio, Silvia Corrado, Marco Bottazzoli, Federico Marchesi, Renata Gili, Pasquale Gianluca Giuri, Davide Gori, Paolo Manzoni

**Affiliations:** 1AUSL–IRCCS di Reggio Emilia, Servizio di Prevenzione e Sicurezza Negli Ambienti di Lavoro (SPSAL), Local Health Unit of Reggio Emilia, 42122 Reggio Emilia, Italy; 2Infectious and Tropical Diseases Unit, Department of Health Promotion, Mother and Child Care, Internal Medicine and Medical Specialties, “G D’Alessandro”, University of Palermo, AOUP P. Giaccone, 90127 Palermo, Italy; antonio.cascio03@unipa.it; 3ASST Rhodense, Dipartimento Della Donna e Area Materno-Infantile, UOC Pediatria, 20024 Milano, Italy; 4Department of Otorhinolaryngology, APSS Trento, 38122 Trento, Italy; 5Department of Medicine and Surgery, University of Parma, 43126 Parma, Italy; federico.marchesi@unipr.it; 6Department of Prevention, Turin Local Health Authority, 10125 Torino, Italy; 7Department of Medicine and Diagnostics, AUSL di Parma, 43100 Parma, Italy; 8Department of Biomedical and Neuromotor Sciences, University of Bologna, 40126 Bologna, Italy; davide.gori4@unibo.it; 9Department of Public Health and Pediatric Sciences, University of Torino School of Medicine, 10125 Turin, Italy; p.manzoni@unito.it

**Keywords:** RSV, viral pneumonia, monoclonal antibodies, immunization

## Abstract

A systematic review with a meta-analysis was performed to gather available evidence on the effectiveness of monoclonal antibody nirsevimab in the prevention of lower respiratory tract diseases (LRTDs) due to respiratory syncytial virus (RSV) in children and newborns (CRD42024540669). Studies reporting on real-world experience and randomized controlled trials (RCTs) were searched for in three databases (PubMed, Embase, and Scopus) until 1 May 2024. Our analysis included five RCTs, seven real-world reports, and one official report from the health authorities. Due to the cross-reporting of RCTs and the inclusion of multiple series in a single study, the meta-analysis was performed on 45,238 infants from 19 series. The meta-analysis documented a pooled immunization efficacy of 88.40% (95% confidence interval (95% CI) from 84.70 to 91.21) on the occurrence of hospital admission due to RSV, with moderate heterogeneity (I^2^ 24.3%, 95% CI 0.0 to 56.6). Immunization efficacy decreased with the overall length of the observation time (Spearman’s r = −0.546, *p* = 0.016), and the risk of breakthrough infections was substantially greater in studies with observation times ≥150 days compared to studies lasting <150 days (risk ratio 2.170, 95% CI 1.860 to 2.532). However, the effect of observation time in meta-regression analysis was conflicting (*β* = 0.001, 95% CI −0.001 to 0.002; *p* = 0.092). In conclusion, the delivery of nirsevimab was quite effective in preventing hospital admissions due to LRTDs. However, further analyses of the whole RSV season are required before tailoring specific public health interventions.

## 1. Introduction

Human respiratory syncytial virus (RSV) is an enveloped, pleomorphic virus of approximately 150 nm (range 120–300 nm) [1] belonging to the *orthopneumovirus* genus of the Pneumoviridae family in the mononegavirales order [2]. Mononegavirales are RNA viruses that have a filamentous single-stranded negative-sense genome. The RSV genome (15 to 16 kb) [3,4,5,6] encodes for a total of 11 proteins [1,7,8,9], including the three main surface antigens: the surface glycoprotein (G, the attachment protein), small hydrophobic (SH) protein, and fusion (F) protein [1,10]. The F protein is also the main pathogenetic factor of RSV as it mediates the relatively complicated invasion process [1,10,11] and significantly contributes to the immune escape strategy of RSV [1,8,10,11]. After the docking of RSV, which is mainly mediated by the interaction of the G protein with CX3C chemokine receptor 1 (CX3CR1) and heparan sulphate proteoglycans (HSPG), F interacts with nucleolin (NCL), epidermal growth factor (EGFR), the receptor of insulin-like growth factor 1 (IGF1R), and intercellular adhesion molecule 1 (ICAM1), enabling the fusion of the host and viral plasma membranes that, in turn, guarantees the passage of viral RNA into the host cells. [1,2,10,12]. During the attachment, the F protein transitions from a pre-fusion (preF) to post-fusion (postF) conformation, eliciting neutralizing antibodies (NA) that target both preF or postF protein [7,13], as well as the cytotoxic T-lymphocyte response, therefore being a suitable target for vaccine development and preventive immunization strategies with monoclonal antibodies (mAb) [13,14,15,16,17]. 

Respiratory epithelia are characterized by the constitutive expression of NCL and CX3CR1, to which both acute and chronic inflammatory status add ICAM1 and HSPG [2,18], representing the main target for RSV, which in fact causes a high burden of respiratory syndromes including acute respiratory infections (ARIs), and lower respiratory tract infections (LRTIs) [5,19,20,21,22,23], particularly in infants and small children [3,5,19,24]. A highly contagious and diffusive pathogen [18,25], at the global level, RSV causes approximately 33 million cases of ARIs and LRTIs in infants aged 5 years old or less every year [4,5,26], with high hospitalization rates [27,28,29,30,31] leading to around 3.5 million hospital admissions [3,5]. Even though RSV can result in a high case fatality ratio in children affected by pre-existing comorbidities [4,5,19,21,26,32,33], most of the cases occur in otherwise healthy infants [4,34] and are usually clustered in seasonal epidemics (i.e., “RSV season”) associated with seasonal climate that forces individuals in enclosed spaces, increasing the likelihood for the inter-human spreading of the pathogen [35,36,37,38]. Therefore, the RSV season in the Northern Hemisphere has historically been associated with the winter season, peaking between December and January [4,26], extensively overlapping with other respiratory viruses such as influenza and adenovirus [39,40] and, more recently, with SARS-CoV-2 [41,42], as well as with the hot, humid, and rainy climates of the summer season in tropical countries [36,37,43,44].

Despite the high disease burden associated with RSV, available options for preventing incident cases of RSV have been limited [18,45,46]. Until recently, RSV candidate vaccines ultimately failed to achieve the targeted vaccine efficacy (VE) of 70% against confirmed severe RSV disease over at least one year post vaccination [47,48]. In fact, only the mAb palivizumab has been licensed and extensively used in real-world settings as a preventative treatment [49,50,51,52]. Palivizumab is an F-protein-targeting mAb: despite its proven efficacy, because of the direct and indirect costs due to the need for several monthly shots (up to 5 during the RSV season) [52,53], palivizumab has only been indicated for a small subset of infants [52,54], which, more precisely, include the following: (1) children who are born at 35 weeks of gestation or less and less than 6 months of age at the onset of the RSV season; (2) children who are less than 2 years of age and required treatment for bronchopulmonary dysplasia within the last 6 months; (3) children who are less than 2 years of age and have hemodynamically significant congenital heart disease [49,50,52,55].

This unsatisfactory landscape has recently been changed by the approval of nirsevimab (Beyfortus, Astrazeneca [Södertälje, Sweden] and Sanofi [Gentilly, France]) [18,56,57], a long-acting mAb that can be delivered as a single dose for the whole of RSV season. Nirsevimab was initially approved by the European Medicine Agency (EMA) in October 2022, before then being authorized by the UK (9 November 2022), Canada (19 April 2023), and, eventually, the USA Food and Drug Administration (FDA) (17 July 2023) [58,59]. Even though the introduction of nirsevimab in the immunization calendars has been hindered by the limited availability of the mAb, as well as by its complicated placement in the vaccination schedules of newborns and infants [58,60], since the inception of the winter season 2023–2024, several real-world experiences have been reported from the Northern Hemisphere [61,62]. This study was, therefore, designed to systematically evaluate the efficacy and provide early estimates on the effectiveness of nirservimab in infants and children in terms of avoiding hospital admissions due to RSV-associated LRTIs, possibly providing guidance for medical and public health professionals.

## 2. Materials and Methods

### 2.1. Research Concept

The outline of the present systematic review with meta-analysis was designed in accordance with the “Preferred Reporting Items for Systematic Reviews and Meta-Analysis” (PRISMA) statement [63,64] (see the PRISMA checklist in Appendix A) and registered on the PROSPERO database (https://www.crd.york.ac.uk/prospero/; accessed on 11 May 2024) with the progressive registration number CRD42024540669 [64].

Research concepts have been defined through the “PECO” strategy (i.e., Patient/Population/Problem; Exposure; Control/Comparator; Outcome) [65,66]. As summarized in Table 1, the main research question of the present study concerned whether individuals aged less than 2 years old who had received at least one dose of nirsevimab (P), upon being exposed to RSV infection during the following RSV season (E), had different rates of occurrence of LRTIs associated with RSV infection (O) compared to children of the same age group that had not been immunized with either nirsevimab or palivizumab (placebo) (C).

### 2.2. Research Strategy

Three databases (PubMed; EMBASE; and Scopus) were specifically searched from inception until 1 May 2024. Moreover, the official websites of national health authorities of EU countries, the USA, and Canada were searched for any report consistent with the research concept. The search strategy was based on the blueprint originally recommended by Turalde-Mapili et al. [67] for their similar study performed in 2023, and it is summarized in Table A1.

### 2.3. Selection Criteria

In order to be included into the present systematic review with meta-analysis, the retrieved studies should provide data on the following:(1)Immunization with nirsevimab of children aged less than 2 years;(2)Nirsevimab mAb administration (any strategy and settings);(3)Comparison of nirsevimab efficacy with placebo in either a randomized controlled trial (RCT) or real-world settings;(4)Reporting on the occurrence of LRTIs in individuals treated with nirsevimab and placebo with subsequent hospitalization.

The following exclusion criteria were then applied: (1)Immunization of children aged 2 years or more;(2)Secondary studies (i.e., systematic reviews and meta-analyses, letters, editorial comment, case reports);(3)Studies on animals (including non-human primates) or preclinical testing;(4)Outcomes other than clinical efficacy;(5)The full text was not available either through online repositories or through inter-library loan or its main text was written in a language different from English, Italian, German, French, Spanish, or Portuguese;(6)A lack of details about the geographical setting and corresponding timeframe;(7)Reporting on the occurrence of influenza-like illnesses and/or respiratory syndromes other than LRTIs;(8)Reporting on the occurrence of LRTIs that did not include the number of cases eventually admitted to the hospital settings because of respiratory syndrome;(9)Methods other than Real-Time Quantitative Polymerase Chain Reaction (RT-qPCR) or non-RT-qPCR Nucleic Acid Amplification Tests (NAATs) were applied for the laboratory diagnosis of RSV infection.

When the retrieved studies included cross-published and/or duplicated data, only the most recent publication was included. When feasible, duplicated data were removed from both qualitative and quantitative analyses.

### 2.4. Selection Criteria

Articles fulfilling the inclusion criteria and not fulfilling in the exclusion criteria were title- and abstract-screened to ascertain their relevance to the research question [64,68]. Only articles that, even during abstract screening, were considered consistent with the research strategy were retained and independently full-text-screened by two investigators (SC, MB). Cases of disagreements were primarily discussed between investigators, and when consensus was not reached, the chief investigator (MR) was consulted as a third arbiter.

### 2.5. Data Extraction

Full-text-screened studies and their Appendix A (where available) were assessed in order to retrieve the following data:

(a) Bibliographic characteristics of the study: The first author’s name, year of publication, and registration number for RCTs;

(b) Characteristics of the clinical study: The sample size of the study groups (recipients of nirsevimab and placebo), baseline data (demographics and comorbidities), number of countries involved in the study, and timeframe of the study. 

(c) Outcome data: Definition of primary and secondary outcomes and case definition. The main outcome encompassed the reported hospital admissions due to RSV-associated LRTIs. 

For studies reporting on several centers and/or patient groups, data were reported as individual series where feasible.

### 2.6. Quality Assessment (Risk of Bias)

The risk of bias (ROB) (i.e., the likelihood that any feature from the design or conduct of a study will lead to misleading results) [69,70,71] was assessed by means of the ROB tool from the National Toxicology Program (NTP)’s Office of Health Assessment and Translation (OHAT) (now the Health Assessment and Translation (HAT) group) [71,72]. OHAT ROB provides a 4-point scale rating (from “definitely low”, “probably low”, and “probably high” to “definitely high”) on the following potential sources of bias: participant selection (D1), confounding factors (D2), attrition/exclusion (D3), detection (D4), and selective reporting (D5), as well as other sources of bias (D6). OHAT ROB was prioritized over other instruments for ROB assessment, as it does not provide an overall rating for each study, nor does it require that studies affected by a certain degree of ROB be removed from the pooled analyses [72].

### 2.7. Data Analysis

#### 2.7.1. Immunization Efficacy

As a preliminary step, the point prevalence of RSV infections among immunized and non-immunized individuals was calculated in percent values. Similarly, individual estimates for the efficacy of nirsevimab were calculated for each study. The efficacy of nirsevimab (immunization efficacy, IE) was calculated in analogy with VE as the percentage reduction in disease cases in a treatment group compared to the non-treated (i.e., placebo) group. More precisely, in the present study, the IE was calculated based on the efficacy in avoiding hospital admissions due to RSV-associated LRTI cases. Mathematically, IE can, therefore, be defined as follows:IE = (1 − Relative Risk) × 100%(1)

#### 2.7.2. Meta-Analysis

Pooled estimates of RSV prevalence and IE were calculated through a random-effect model (REM) meta-analysis of retrieved studies and reported as point estimates with their 95% confidence intervals (95% CI). In REM, the effect of the categorical variables is defined as varying across the levels of the variable. In other words, as random effects account for variability and differences between different entities or subjects within a larger group, the REM was considered more effective at dealing with the genuine differences underlying the results of the studies, or heterogeneity [73,74]. Heterogeneity, in turn, can be defined as the inconsistency of the effect between the included studies, or the percentage of total variation across included studies likely due to actual differences rather than chance [69]. Heterogeneity can be quantified in percent values by means of I^2^ statistic. To fit the aims of the following studies, heterogeneity was considered low for I^2^ values ranging from 0 to 25%, moderate for I^2^ values ranging from 26% to 50%, and substantial for I^2^ values ≥50%. In order to cope with the presumptively reduced number of studies included in the meta-analyses, 95% CIs of I^2^ estimates were calculated and reported [69]. 

#### 2.7.3. Sensitivity Analysis

In order to evaluate the effect of each study on the pooled estimates, a sensitivity analysis (i.e., a study of how the uncertainty in the output of a mathematical model or system can be apportioned to different sources of uncertainty in its inputs) was performed by excluding one study at a time. 

#### 2.7.4. Analysis of Publication Bias

Publication bias was initially assessed through the visual analysis of the asymmetry of funnel plots. Contours of statistical significance were added in order to aid the visual interpretation of the plots. The asymmetry of funnel plot outcomes with three or more included studies was then assessed by means of Egger’s test [64,75]. Small-study bias was eventually assessed by means of radial plots. A *p* value < 0.05 was considered statistically significant for both publication and small-study bias.

#### 2.7.5. Software

All calculations were performed by means of R (version 4.3.1) [76] and Rstudio (version 2023.06.0 Build 421; Rstudio, PBC; Boston, MA, USA) software by means of the packages meta (version 7.0) and fmsb (version 0.7.5). A Prisma2020 flow diagram was designed by means of the PRISMA2020 package [77].

## 3. Results

### 3.1. Characteristics of Retrieved Studies

As summarized in Figure 1 and detailed in Table A1, a total of 401 entries were retrieved: 60 studies from PubMed (14.96%), 145 from Scopus (36.16%), and 196 from EMBASE (48.88%). A total of 168 entries were duplicated across the searched databases (41.90%), before being removed from the screening, which eventually included a total of 233 titles and abstracts. Of them, 121 were excluded (30.17%), while 112 entries were sought for retrieval (27.93%) and then full-text-assessed. Eventually, 102 studies were removed as being inconsistent with inclusion criteria (55, 13.72%) or being nonoriginal studies (46, 11.47%), leading to the inclusion of 10 reports (2.49%) [59,61,78,79,80,81,82,83,84,85]. Moreover, through the analysis of the official websites of the public health authorities, three further entries were identified as consistent with the inclusion criteria [86,87,88]. 

The individual characteristics of the included studies are summarized in Table 2. In brief, the pooled sample included five reports from RCTs from the clinical research on nirsevimab [78,79,80,81,82], seven studies on real-world experiences [59,61,83,84,85,87,88], and a further report from the local health authority of Galicia [86]. The observation times of the included studies (i.e., the time from the delivery of nirsevimab to the end of the registration of incident LRTIs) ranged between 57 days and 805 days, with four studies including an observation time < 150 days [83,84,85,88]. However, only the report from Dagan et al. [82] included data on the follow-up season following the initial delivery of nirsevimab. Six out of seven of real-world studies [59,83,84,85,87,88] and the official report from NIRSEGAL study [86] were conducted in Western Europe, with the earliest recruitment from 25 September 2023 [59]. More precisely, 4 studies and 13 case series were from Spain [59,84,85,86], while 1 study each from France [87], Luxembourg [83], Italy [88], and the USA [61]. Sample size ranged from a minimum of 288 infants for a study that only included subjects hospitalized because of RSV-LRTIs in pediatric intensive care units (PICU) to 10,259 patients from the real-world report of Ares-Gómez et al. [59] and 15,676 patients from the real-world report of López-Lacort et al. [84]. The reported hospital admission rate for LRTIs in infants treated with nirsevimab ranged from 0 [88] to 1.96% [83], while it ranged between 1.61% [79,80,82] and 63.59% [61], and even 87.40% [87], for infants not treated with nirservimab. Based on the reported data, corresponding point estimates on IE for nirsevimab ranged from 62.6% (−7.3 to 86.9), reported by the early stages of RCT NCT03979313 [79,80,82], to 98.4% (72.9 to 99.9), reported by the study of Consolati et al. [88]. 

As stated in Table A2, among retrieved RCTs, the treatment of either cases or controls with palivizumab was documented only by Simoes et al. [79], as all controls (n. 290) actually received a full conventional immunization course. On the other hand, all sampled children were not exposed to a maternal immunization strategy. When dealing with real-world studies, while the studies by Lopez-Lacort et al. [84], Ernst et al. [83], and Ezpeleta et al. [85] did not document the actual exposure of sampled infants to both palivizumab and maternal immunization, the studies by Consolati et al. [88], Paireau et al. [87], and Moline et al. [61] deliberately excluded from their reports all cases managed with strategies alternative to nirsevimab. Finally, the report from the NIRSEGAL study [59,86] did not include any case managed with palivizumab, and no information was provided on maternal immunization.

### 3.2. Risk of Bias Assessment

The quality of the included studies is summarized in Figure 2, while their individual assessment is provided as the Table A3. Nearly all studies were characterized by high quality and a low risk of bias. The notable exception was represented by the report by Paireau et al. [87], the only report still in pre-print status at the time of this review. In fact, the authors implemented a design quite inconsistent with those of other real-world studies, as they reported on hospitalizations due to RSV-associated LRTIs, not providing the total number of doses delivered in the general population or in a specific group of infants. Moreover, the study included cases from PICU, compared to the general population used in other reports. Another partial exception was represented by the reports on the NIRSEGAL study group sourced from local health authorities, as the total number of hospital admissions due to RSV-associated LRTIs in non-treated children was not provided [86]. 

### 3.3. Quantative Analysis

#### 3.3.1. Removal of Studies

Because of its design, as it focused on incident cases of LRTIs due to RSV in PICU, the report from Paireau et al. [87] was considered inconsistent with other reports, both RCTs and observational studies, therefore being removed from quantitative analysis. Cases from duplicated series [59,79,80,82,86] were similarly removed. The final sample, therefore, included a total of 45,238 infants (Table 3). 

#### 3.3.2. Descriptive Analysis

Overall, 43,294 (95.70%) infants were reported in the first season after the delivery of nirsevimab: of them, 33,884 had been immunized (78.35%), while 9365 (21.65%) were placebos (RCTs) or controls (real-world studies) not immunized with nirsevimab. Of them, 290 were otherwise immunized with palivizumab (3.10%). Data for the follow-up season only included a total of 1944 further cases (4.30%), with a total of 6 cases of hospitalizations due to RSV-associated LRTIs (hospital admission rates of 0.15% and 0.31% for subjects immunized and not immunized, respectively) [82].

In detail, the occurrence of hospital admissions due to RSV-associated LRTIs was equal to 0.42% among infants treated with nirsevimab, compared to 7.22% among non-treated subjects, with a substantially reduced risk ratio (RR) of 0.058, with a 95% CI 0.049 to 0.070. Interestingly, when the sampled infants were assessed based on the design of the studies, the rate of hospitalization due to RSV-associated LRTIs was 2.26% in real-world studies compared to 1.08% in RCTs (RR 2.099, 95% CI 1.756 to 2.510). Similarly, an increased occurrence of hospitalization due to RSV-associated LRTIs was identified for infants sampled from studies with an observation period of 150 days or more (RR 2.170, 95% CI 1.860 to 2.532). When dealing with the timeframe of the study, no substantial differences were identified for having been a study completed before (hospitalization rate 1.53%) or after the inception of COVID-19 pandemic (1.93%, RR 1.263, 95% CI 0.969 to 1.646). Eventually, using the rate of hospitalizations due to RSV-associated LRTIs in RCTs as the reference group, studies from Spain exhibited a substantially reduced risk (RR 0.603, 95% CI 0.484 to 0.751), while those from other EU countries (RR 3.876, 95% CI 2.980 to 5.043) and, in particular, the single USA study included in the analysis (RR 54.887, 95% CI 46.132 to 65.303) were associated with an increased risk of hospitalization. 

As shown in Figure 3a, no substantial correlation was found between observation time and hospitalization rates for infants treated and not treated with nirsevimab (Spearman’s r = 0.340, 95% CI −0.149 to 0.696, and *p* = 0.154 and r = −0.076, 95% CI −0.523 to 0.404, and *p* = 0.758, respectively), while point estimates for hospital admission rates were significantly correlated (Spearman’s r = −0.546, 95% CI −0.807 to −0.108, and *p* = 0.016). Eventually, immunization efficacy per single study and observation time were negatively correlated (Spearman’s r = −0.546, 95% CI −0.807 to −0.108, *p* = 0.016).

#### 3.3.3. Meta Analysis

Prevalence estimates for hospitalizations due to RSV-associated LRTIs among children previously immunized by means of nirsevimab are reported in Figure 4. In brief, a pooled estimate of 0.42% (95% CI 0.26 to 0.68) was calculated: no substantial differences were identified between RCTs (0.43%, 95% CI 0.28 to 0.68) and studies reporting on the real-world experience (0.40%, 95% CI 0.20 to 0.78; chi-squared test 0.05, *p* = 0.830). Prevalence estimates were affected by substantial heterogeneity (I^2^ 86.1%, 95% CI 79.7 to 90.5; tau^2^ = 0.912, Q 129.70, *p* < 0.001), mostly due to real-world studies (89% vs. 42.0% for RCTs).

A corresponding forest plot for hospitalizations due to RSV-associated LRTIs is provided as Annex Figure A1. A total of 9365 children not immunized with nirsevimab were pooled as a sample, with a cumulative prevalence of 4.25% (95% CI 2.42 to 7.38) calculated by means of REM, including a prevalence of 2.45% (95% CI 1.61 to 10.34) for RCTs compared to 4.25% (95% CI 2.42 to 7.38) for observational studies from real-world settings, with no substantial differences between groups (chi-squared test 2.98, *p* = 0.084). Also, estimates for non-treated infants were affected by substantial heterogeneity (I^2^ 98.8%, 95% CI 98.5 to 99.0; tau^2^ = 1.481, Q = 2.98, *p* < 0.001), even when considered by subgroups (I^2^ 84% for real-world observational studies vs. 92% for RCTs).

Pooled estimates on IE are reported in Figure 5. As shown, a cumulative estimate of 88.4% (95% CI 84.7 to 91.2) was calculated with substantial differences between RCTs (81.0%, 95% CI 71.5 to 87.3) and real-world observational studies (90.5%, 95% CI 87.1 to 92.9; chi-squared test 7.20, *p* value = 0.007).

Pooled estimates were affected by moderate heterogeneity (I^2^ 24.3%, 95% CI 0.0 to 56.6; tau^2^ = 0.105, Q = 23.79, *p* = 0.162), even though the individual subgroups were otherwise exempt (I^2^ 0% for RCTs, I^2^ 3% for real-world observational studies).

In order to assess whether observation time had any effect on the pooled estimates, a metaregression was implemented, with the subsequent calculation of the corresponding bubble plot. In the bubble plot, the observed outcomes (in this case, immunization efficacy) of the individual studies are plotted against a quantitative predictor (in this case, the observation time) and the size of the “bubble” is proportional to the weight that the studies received in the analysis (i.e., studies receiving more weight have larger points) (Figure 6). Eventually, observation time was characterized as a non-significant explanatory variable for IE (*β* = 0.001, 95% CI −0.001 to 0.002; *p* = 0.092), and the analyses were affected by low heterogeneity (I^2^ = 19.0%, tau^2^ = 0.055).

### 3.4. Sensitivity Analysis

Sensitivity analysis was performed through the removal from pooled estimates of hospital admission rates and the IE of a single study at a time. The resulting pooled estimates are reported in Figure A2, Figure A3, Figure A4 and Figure A5. Regarding hospital admission rates (Figure A3 and Figure A4), the removal of single studies did not affect either estimates or heterogeneity. Contrarily, the removal of the series from the Hospital Clinico Universitario Virgen de la Arrixaca from the study of López-Lacort et al. [84] (4631 patients, 33 cases of hospital admissions) led to a substantial reduction in the point value of I^2^, seemingly estimated as 0%.

### 3.5. Publication Bias

Publication bias was preliminarily assessed via the calculation and visual inspection of funnel plots. According to the PRISMA guidelines [64,68,77], the sample size was plotted against the effect size otherwise reported in the plot. As the size of the sample increases, individual estimates of the effect are likely to converge around the true underlying estimate [63,68,77]. Funnel plots for hospital admission rates are reported in Figure A5, while the funnel plot for IE estimates is provided in Figure 7a. Estimates were scattered across the plots and highly symmetrical, ruling out the presence of publication bias due to the high share of lower prevision studies. In fact, Egger’s test for funnel plot asymmetry was similarly not significant for estimates of admission rates for subjects treated with nirsevimab (t = −1.06, *p* value = 0.305, tau^2^ = 7.218) and IE (t = −0.54, *p* value = 0.597, tau^2^ = 1.376), while it confirmed the publication bias for prevalence studies on the admission rates for non-immunized infants (t = −2.87, *p*-value = 0.011, tau^2^ = 58.632).

Even in radial plots for the IE and hospital admissions (Figure 7b, Figure A7a), individual estimates were substantially scattered across the regression line when dealing with rates for subjects treated with nirsevimab, ruling out the small-study effect that was otherwise noticeable with estimates for prevalence in non-immunized infants (Figure A6b).

## 4. Discussion

### 4.1. Summary of Main Findings

In this systematic review with a meta-analysis, we were able to retrieve a total of 13 studies on the use of nirsevimab for preventing hospitalizations due to LRTIs associated with RSV infection [59,61,78,79,80,81,82,83,84,85,86,87,88], including 7 longitudinal studies [59,61,83,84,85,86,88], 5 RCTs [78,79,80,81,82], and 1 observational case–control study on PICU admissions [87], that were, therefore, not included in pooled analyses. As reports on the RCT NCT03979313 were cross-reported [79,80,82], and, similarly, the NIRSEGAL study was otherwise included in the paper by Ares-Gómez et al. [59], while the study by López-Lacort et al. [84] eventually included nine series, analyses were performed on a total of 19 patient series [59,61,78,79,82,83,84,85,88]. All papers included in this review and, in particular, in the meta-analysis were of high or even very high quality, and the risk of literature bias was, hence, very low. A pooled hospitalization rate for RSV-associated LRTIs was calculated as 0.42% (95% CI 0.26 to 0.68) for infants immunized with nirsevimab and 4.25% (95% CI 2.42 to 7.38) for those not immunized. A corresponding IE of 88.4% (95% CI 84.7 to 91.2) was, therefore, calculated, being significantly greater in real-world studies than in RCTs (90.5%, 95% CI 87.1 to 92.9 vs. 81.0%, 95% CI 71.5 to 87.3 in RCTs; *p* value = 0.007), and sensitivity analysis did not suggest any substantial role for outliers in the pooling of total estimates. 

### 4.2. Generalizability and Implications for Daily Practice

Efficacy (i.e., the degree to which an intervention prevents a certain outcome) is calculated in RCTs under ideal and controlled circumstances, and even though a certain intervention with a documented high efficacy (e.g., COVID-19 mRNA vaccines against original Wuhan strains of SARS-CoV-2) would be expected to be also high performing (i.e., effective) in real-world settings, factors such as how the intervention is delivered underlying and heterogenous clinical conditions and the actual occurrence of the targeted condition (in this case, the RSV epidemiology) can reduce how effective the intervention is at preventing the disease [89]. According to our data, the actual efficacy of nirsevimab in avoiding hospitalizations due to RSV-associated LRTIs was comparable to its real-world effectiveness, stressing how nirsevimab’s potential impact on the management of RSV disease burden should be taken into consideration not only from a clinical point of view but also from a public health perspective [59,88]. While palivizumab is highly effective in avoiding severe respiratory syndromes associated with RSV infections in highly selected patient groups [50,52,55,90], nirsevimab potentially provides a preventive option to be delivered to the whole of the infant population, with a likely impact on the RSV-associated disease burden [57,62,91]. An often-overlocked and even forgotten disease [92,93], particularly among adults and the elderly [45,46,94], RSV has recently emerged from the pandemic as a generally acknowledged cause of medical consultations and hospitalization [25,93]. The increasing notoriousness of RSV in the general population likely has several causes. On one hand, during the first months of COVID-19 pandemic, the prevention of RSV infections benefited from non-pharmaceutical interventions for respiratory pathogens primarily implemented against SARS-CoV-2 [95,96,97], but their removal led to an unprecedented increase in incidence rates reported at the global level in the following years [98,99,100,101]. On the other hand, several preventative options have ultimately been made available to medical professionals and potential end users, including parents and caregivers, such as nirsevimab and two effective subunit vaccines for older adults: Abrysvo from Pfizer Inc. (Pfizer Europe MA EEIG, Brussels, Belgium) and Arexvy from GlaxoSmithKline LLC (GlaxoSmithKline Biologicals SA, Rixensart, Belgium) [48,102,103]. Unfortunately, both vaccines cannot be used in infants and children [102,104,105], and even though both vaccines have been designed to be delivered in pregnant women, only Abrysvo has been approved by the Food and Drug Administration (FDA) and the European Medicine Agency (EMA) for maternal use (32 to 36 weeks gestational age of pregnancy for the FDA, 24 to 36 weeks for the EMA) [106]. Even though a recent meta-analysis has stressed the potential efficacy of maternal immunization for reducing hospitalizations in offspring (RR 0.50, 95% CI 0.31 to 0.82) [107], safety issues have been raised for Arexvy, particularly regarding preterm births (relative risk 1.37, 95% CI 1.08 to 1.74, *p* = 0.001), forcing GSK to suspend the recruitment of new cases during a phase 3 study (RCT NCT04605159) [108]. In such a setting, nirsevimab could represent a likely option for guaranteeing effective protection against LRTIs in all children, not only for children currently benefiting from the preventive treatment with palivizumab.

Still, several caveats should be mentioned. According to its design and preliminary studies, nirsevimab should guarantee around 150 days of protection against RSV [79,80,109], being particularly suitable for managing a full RSV season with a single shot compared to the monthly treatment required by palivizumab [78]. Unfortunately, several real-world studies were completed well before the 150th days of the observation period [83,84,85,87,88], and not coincidentally, studies with an observation period longer than 150 days (i.e., the coverage guaranteed by a single dose of nirsevimab) were characterized by higher rate of hospitalizations due to RSV-associated LRTIs (RR 2.170, 95% CI 1.860 to 2.532) compared to the former ones. Similarly, the individual estimates for IE were negatively correlated with the length of the observation period (Spearman’s r = −0.546, *p* = 0.016), i.e., the longer the observation time, the lower the actual estimate for the IE. Still, it should be stressed that meta-regression analysis did not confirm the actual role of observation time. 

Another criticism of the generalizability of collected data is represented by the immunization strategy implemented in most real-world studies. For logical, ethical reasons, in longitudinal studies, all newborns from a certain geographical area were the potential recipients of nirsevimab [59,61,83,84,85,86,88]. As a consequence, non-treated individuals were children whose parents were either unwilling or unable to participate in this preventive intervention. In strict analogy with vaccine hesitancy, we cannot rule out that non-treated children belonged to families characterized by lower health literacy, lower socio-economic status, and some degree of social marginalization [110,111,112], which, in turn, may increase the likelihood of respiratory infections [113,114]. On the other hand, because of the current epidemiology of RSV, particularly in European countries, and the different immunization strategies implemented by local health authorities, the comparisons with previous RSV seasons would be of limited significance. While the delivery of nirsevimab among newborns and infants led to substantial reductions in the total number of hospitalizations (Table A4) [83,86,88], ranging from −30.08% [83] to −77.05% [88], recently reported Spanish estimates on hospitalizations due to RSV-associated LRTIs hint at a very high rate for the RSV season in 2022 compared not only to the pandemic seasons 2021 and 2020 (606 vs. 395 and 247, respectively) but also to the pre-pandemic years 2016 (n = 551), 2017 (n = 451), and 2018 (n = 366), only being comparable to the peak year 2019 (n = 642) [115]. 

Finally, the IE was calculated only in longitudinal studies: even though the report from Moline et al. [61] was primarily meant to be a case–control study, we were able to pool reported data by means of its mixed design. On the contrary, the case–control report of Paireau et al. [87] was removed from pooled estimates. For both studies, an alternative and appropriate calculation of actual IE could be based on the odds ratio rather than relative risk. However, as reported in Table A5, both studies would be still associated with a high IE, i.e., 74.6% (50.7 to 86.9) for Paireau et al. [87] and 81.0% (55.1 to 92.0) for Moline et al. [61].

### 4.3. Limitations and Implications for Future Studies

Despite its potential significance in providing an early estimate of IE of nirsevimab against hospital admissions due to RSV-associated LRTIs, we must acknowledge several limitations of the present study.

As a preliminary step, the strict search strategy on this relatively new topic (i.e., a new mAb such as nirsevimab, commercialized in EU only since the winter season 2023) has likely guaranteed the good or very good quality of retrieved studies, but it similarly resulted in a small number of collected reports. This very same shortcoming has been identified in some similarly designed systematic review with a meta-analysis on the prevention of RSV infections [48,107,116] and will likely affect all secondary studies on RSV prevention in the upcoming years until the more extensive use of new vaccines and mAb occurs in real-world settings. 

Second, despite the high or even very high quality of the retrieved studies, diverse immunization and reporting strategies, particularly from real-world studies, have reasonably introduced some degree of heterogeneity into the results, as suggested by pooled estimates on hospital admission rates, potentially affecting the collected results. For instance, while RCTs likely did not include any patient exposed to a maternal immunization strategy and deliberately excluded cases managed with palivizumab, with the notable exception of the MEDLEY subgroup of the study by Simoes et al. [79], reporting strategies from real-world studies are a lot more heterogeneous. On one hand, since the beginning of nirsevimab immunization, the delivery of palivizumab has been discontinued in the Spanish region of Galicia [59], but no information is provided from other Spanish regions. On the other hand, the studies of Consolati et al. [88], Moline et al. [61], and Paireau et al. [87] deliberately removed from their estimates infants immunized with palivizumab. Similarly, the exposure of sampled children to maternal immunization is ruled out for the aforementioned studies, while no information is provided by all of Spanish studies [59,84,85]. Moreover, because of its design, we were unable to include in pooled estimates the study of Paireau et al. [87]. From both a clinical and public health point of view, it was particularly frustrating as this study documented the efficacy of nirservimab in a specific and quite significant subset of patients, i.e., admission to PICU [87,117,118], the prevention of which would lead to a substantial reduction in the eventual RSV-associated disease burden in terms of both morbidity and direct and indirect costs. However, the point estimate of presumptive IE was not only in line with the general estimate but also highly comparable with that from the study of Moline et al. [61] when IE was estimated, considering the case–control design of the aforementioned studies. 

Third, it is very important to stress how RCTs included in this systematic review with meta-analyses were substantially influenced by the COVID-19 pandemic in both a direct and an indirect way. According to the preliminary design of the RCT NCT03979313 (i.e., MELODY study) [80], a full trial size of 3000 participants was envisaged, but only 1500 patients were actually recruited before the World Health Organization declared the COVID-19 pandemic on 11 March 2020. As previously stressed, the implementation of lockdown measures and non-pharmaceutical interventions radically affected the global epidemiology of RSV [97,98,101,119,120], potentially impairing the eventual reliability of subsequent analyses [80,82,121,122]. As a consequence, the report from Hammitt et al. [80] only included a primary analysis, and the report on the full trial size was delayed in post-pandemic settings [82]. The potential consistency of pre- and post-pandemic data could, therefore, be questioned. According to our estimates, while the overall occurrence of hospital admissions due to LRTIs was not significantly higher in post-pandemic settings compared to pre-pandemic studies (RR 1.263, 95% CI 0.969 to 1.646), the risk of hospital admission due to breakthrough infections in children treated with nirservimab from the RCT NCT03979313 was greater in pre-pandemic settings compared to post-pandemic settings (0.60%, 95% CI 0.22 to 1.31 vs. 0.30%, 95% CI 0.06 to 0.86; RR 2.042, 95% CI 0.512 to 8.143). Moreover, the collected evidence included the report from Simoes et al. [79], which, in turn, provided data on the efficacy of nirsevimab from a phase 2b study (i.e., MEDLEY study) [109] that was originally designed for evaluating the safety of this mAb in a subset of infants with a high rate of premature heart or lung disease. Because of the original aim of the MEDLEY study, controls, therefore, received a full prophylaxis with palivizumab rather than placebo. Despite the rigorous design of the original RCT, its eventual consistency with other reports mostly focusing on healthy children could be then questioned.

Fourth, it should be stressed that because of the specific characteristics of RSV infection, a large share of diagnoses may be missed, even among hospitalized patients [48,123]. On one hand, the collection of respiratory tract specimens could fail to provide an appropriate diagnosis in accordance with the stage of the assessed infection [124,125]. Severe cases associated with LRTIs may result in false-negative results when the specimens are collected from the upper airways, as the viral infection is more likely to be active in the lower regions of the respiratory tract [113]. In fact, when provided by the collected study, the proportion of RSV-associated LRTIs among the total number of hospitalized LRTIs (i.e., including both RSV-associated and non-RSV associated cases) ranged between 28.28% (95% CI 19.69 to 38.22) [78] and 100% [85] (Figure A7), suggesting that the actual proportion of RSV-associated cases may have somehow been underestimated, conversely inflating the pooled estimates for the IE. The choice of limiting our analysis on the outcome of hospital admissions due to RSV-associated LRTIs likely limited the potential heterogeneity due to the variable case definition of LRTIs and diagnostic options from RCTs and real-world settings, being otherwise recommended for the appropriate appraisal of RSV vaccines [48,105] Nonetheless, hospital admission rates for RSV-associated respiratory syndromes have been extensively documented as being highly heterogeneous, mostly due to national policies and the availability of healthcare options [19,126,127,128]. As the largest proportion of included samples from real-world settings were provided by studies from the Spanish settings [115], a cautious approach when generalizing our estimates, not only at global level but also at the regional (i.e., European) level, is, therefore, required.

Finally, when addressing the potential use of nirsevimab in real-world settings, the potential emergence of resistance should be considered, particularly in the light of the development of future vaccines. In fact, while the binding site of nirsevimab (site ø) is highly conserved due to its functional requirements [129], there is considerable evidence that even major variant substitutions within the nirsevimab binding site do not reduce the potency of this mAb [122,129]. However, as preF and, in particular, its site ø are a shared target for vaccines such as RSVpreF and RSVPreF3, as well as for mRNA-1345 [130], the extensive reliance on nirsevimab for protecting infants and newborns should be associated with the accurate and continuous monitoring of new variants potentially emerging due to its selective pressure.

## 5. Conclusions

The RSV immunization of infants and children with nirsevimab has been proven to be effective at preventing hospitalizations due to LRTIs. However, due to the limited availability of complete and fully comparable follow-up for all retrieved RCT studies, the limits to the generalizability of data from real-world experience, and the potential underestimation of incident cases of LRTIs associated with RSV, further and improved reporting from ongoing real-world experiences is still recommended in order to properly address the cost-effectiveness profile of this innovative preventative option.

## Figures and Tables

**Figure 1 vaccines-12-00640-f001:**
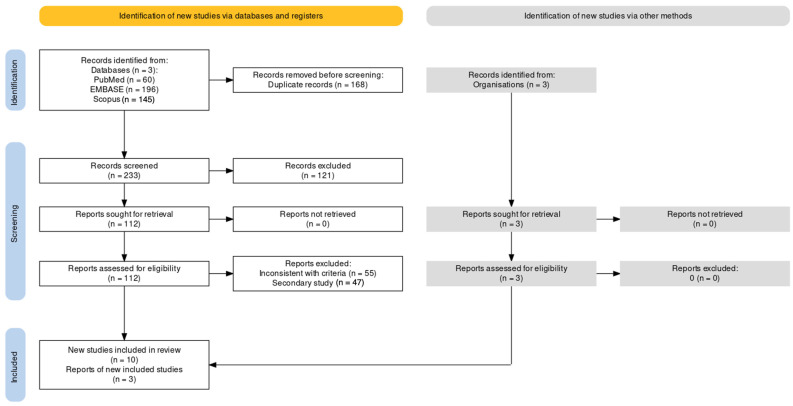
Flowchart of included studies.

**Figure 2 vaccines-12-00640-f002:**
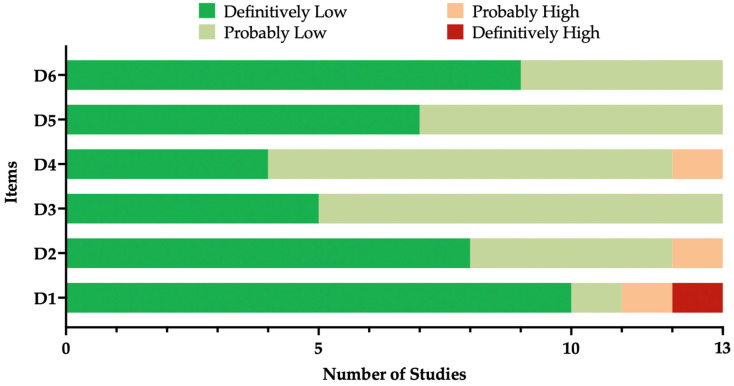
Summary of the risk of bias (ROB) estimates for observational studies [72,85]. Analyses were performed according to the National Toxicology Program (NTP)’s Office of Health Assessment and Translation (OHAT) handbook and respective ROB tools including all retrieved studies (n = 13).

**Figure 3 vaccines-12-00640-f003:**
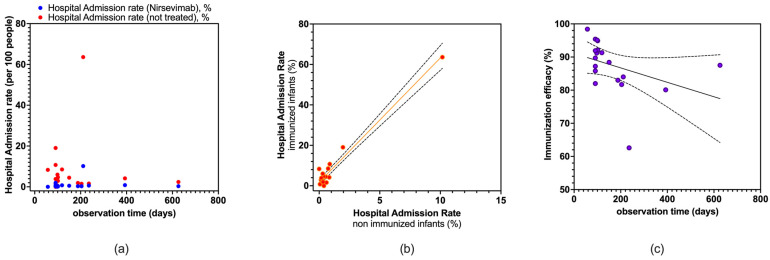
Correlations between the following: (**a**) observation time and hospital admission rate as events per 100 people; (**b**) hospital admission rates among immunized and not immunized infants; (**c**) observation time and point estimates for immunization efficacy per single study. No substantial correlation was found between observation time and hospitalization rates for infants treated and not treated with nirsevimab (Spearman’s r = 0.340, 95% CI −0.149 to 0.696, and *p* = 0.154 and r = −0.076, 95% CI −0.523 to 0.404, and *p* = 0.758, respectively), while a positive correlation was identified between hospital admission rates (Spearman’s r = 0.473, 95% CI 0.009 to 0.769, and *p* = 0.041), and a negative correlation was found for immunization efficacy and observation time (Spearman’s r = −0.546, 95% CI −0.807 to −0.108, and *p* = 0.016) [59,61,78,79,82,83,84,85,88].

**Figure 4 vaccines-12-00640-f004:**
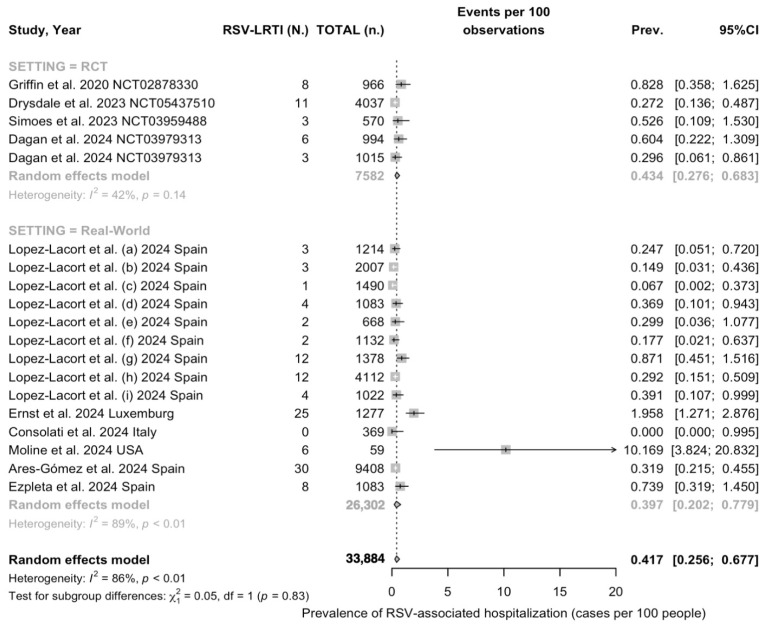
Forest plot for hospitalization rates due to RSV-associated LRTIs among a pooled sample of 33,884 children immunized with nirsevimab. A rate of 0.42% (95% CI 0.26 to 0.68) was calculated for 143 total hospital admissions (I^2^ 86.1%, 95% CI 79.7 to 90.5; tau^2^ = 0.912, Q 129.70, *p* < 0.001), with no substantial differences between randomized controlled trials (RCTs) and real-world experience (chi-squared test 0.05, *p* value = 0.830) [59,61,78,79,82,83,84,85,88].

**Figure 5 vaccines-12-00640-f005:**
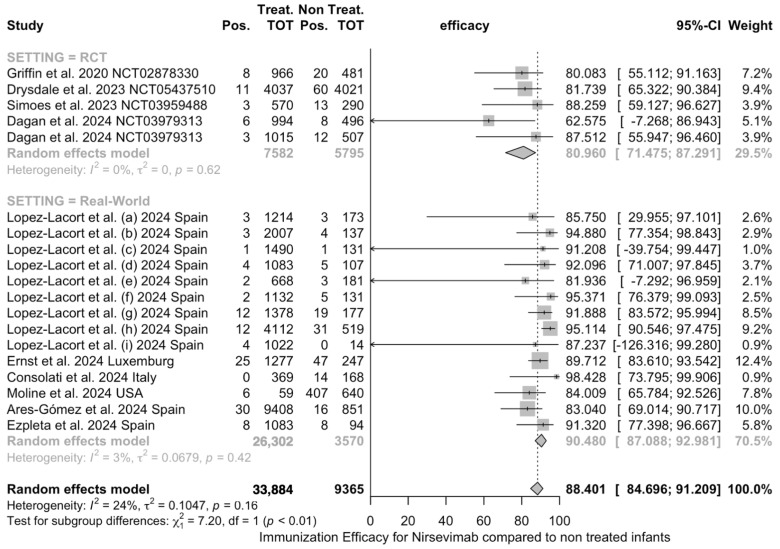
Forest plot for immunization efficacy (IE) of nirsevimab against RSV-associated LRTIs in real-world experience and randomized controlled trials (RCTs). A pooled IE of 88.4% (95% CI 84.7 to 91.2) was calculated (I^2^ 24.3%, 95% CI 0.0 to 56.6; tau^2^ = 0.105, Q = 23.79, *p* = 0.162), with substantial differences between groups (chi-squared test 7.20, *p* value = 0.007) [59,61,78,79,82,83,84,85,88].

**Figure 6 vaccines-12-00640-f006:**
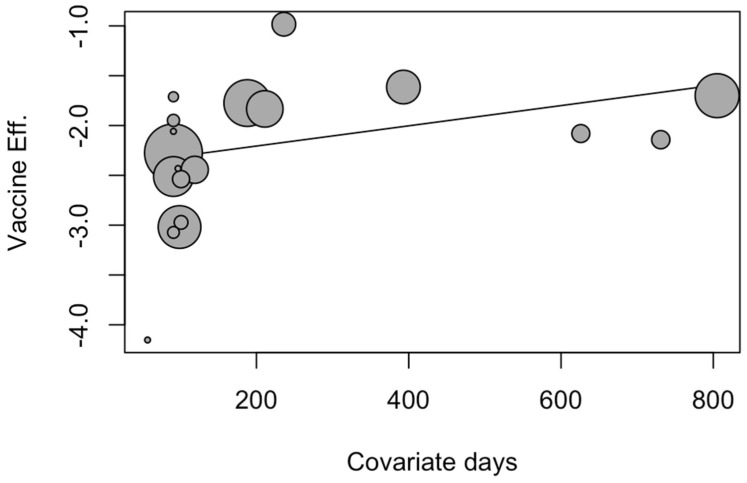
Bubble plot for the metaregression of the included studies by observation days. In the regression model, the observation time was characterized as a non-significant explanatory variable for IE (*β* = 0.001, 95% CI −0.001 to 0.002; *p* = 0.092). Pooled analyses were affected by low heterogeneity (I^2^ = 19.0%, tau^2^ = 0.055) [59,61,78,79,82,83,84,85,88].

**Figure 7 vaccines-12-00640-f007:**
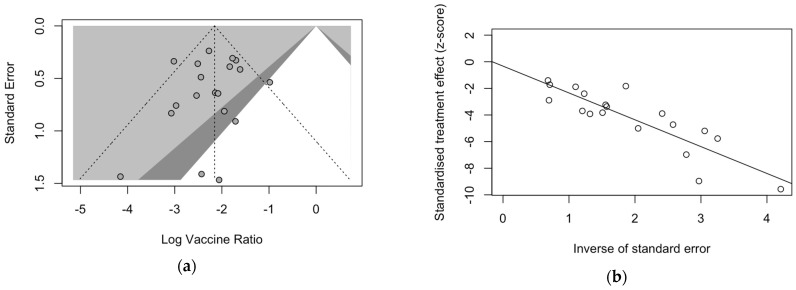
The funnel plot (**a**) and radial plot (**b**) for studies reporting on the immunization efficacy of nirsevimab on the occurrence of hospital admissions due to respiratory syncytial virus (RSV)-associated lower respiratory tract infections. A regression test of funnel plot asymmetry (Egger’s test) ruled out substantial publication bias due to the small-study effect (t = −0.54, df = 17, *p*-value = 0.597, bias estimate: −0.324, SE = 0.601, tau^2^ = 1.376) [59,61,78,79,82,83,84,85,88].

**Table 1 vaccines-12-00640-t001:** PECO worksheet [65,66].

Item	Definition
Population of Interest	Children who had received at least one dose of nirsevimab for the prevention of respiratory syncytial virus infection
Exposure	exposed to RSV infection during the subsequent RSV season
Control/Comparator	Children who had not received nirsevimab or palivizumab for the prevention of respiratory syncytial virus infection (placebo)
Outcome	Occurrence of lower respiratory tract infection

**Table 2 vaccines-12-00640-t002:** Summary of studies included in the systematic review (Note: MC = multicenter study; SC = single-center study; LRTI = lower respiratory tract infection; Obs. = observation; RCT = randomized controlled trial; IE = immunization efficacy).

Study	Year	Settings	Timeframe	Design	Obs. Time (Days)	Total Sample (N.)	Nirsevimab		Controls		IE (95% CI)
							Total (n./N., %)	RSV-Associated LRTI Cases with Hospital Admission (n./n.)	Total (n./N., %)	RSV-Associated LRTI Cases with Hospital Admission (n./n.)	
Griffin et al. [78]	2020	RCT (NCT02878330)	3 November 2016 to 1 December 2017	MC 23 countries	393	1453	966 (66.48%)	8 (0.83%)	481 (33.52%)	20 (4.16%)	80.1% (55.1 to 91.2)
Hammitt et al. [80]	2022	RCT * (NCT03979313)	23 July 2019 to 15 March 2020	MC 20 countries	236	1490	994 (66.71%)	6 (0.60%)	496 (33.29%)	8 (1.61%)	62.6% (−7.3 to 86.9)
Drysdale et al. [81]	2023	RCT (NCT05437510)	8 August 2022 to 28 February 2023	MC France, Germany, UK	204	8058	4037 (50.10%)	11 (0.27%)	4021 (49.90%)	60 (1.49%)	81.7% (65.3 to 90.4)
Simoes et al. [79]	2023	RCT * (NCT03979313)	23 July 2019 to 15 March 2020	MC 20 countries	236	1490	994 (66.71%)	6 (0.60%)	496 (33.29%)	8 (1.61%)	62.6% (−7.3 to 86.9)
		RCT (NCT03959488)	2019–2021	MC 23 countries	>150	860	570 (66.28%)	3 (0.53%)	290 (33.72%)	13 (4.48%)	88.4% (59.8 to 96.7)
López-Laco et al. [84]	2024	Real World	1 October 2023 to 10 January 2024	Spain, MC 8 centers	91 to 101	15,676	14,106 (89.98%)	43 (0.30%)	1570 (10.02%)	52 (3.31%)	92.2% (88.6 to 94.6)
Ernst et al. [83]	2024	Real World	1 October 2023 to 31 December 2023	Luxemburg, SC	91	1524	1277 (83.80%)	25 (1.96%)	247 (16.20%)	47 (19.03%)	89.7% (83.6 to 93.5)
NIRSEGAL Study [86]	2024	Real World (at birth)	1 October 2023 to 3 March 2024	Spain, MC (Galicia)	154	7278	6723 (92.37%)	52 (0.77%)	555 (7.63%)	N.A.	-
		Real World (catch-up)				7354	5824 (79.19%)	37 (0.64%)	1530 (20.81%)	N.A.	-
Dagan et al. [82]	2024	RCT * (NCT03979313) (original study)	23 July 2019 to 15 March 2020	MC 31 countries	236	1490	994 (66.71%)	6 (0.60%)	496 (33.29%)	8 (1.61%)	62.6% (−7.3 to 86.9)
		RCT (NCT03979313) (completed)	23 July 2019 to 9 April 2021		626	1522	1015 (66.69%)	3 (0.30%)	507 (33.21%)	12 (2.37%)	87.5% (55.9 to 96.5)
		RCT (NCT03979313) (second season)				2911	1944 (66.78%)	3 (0.15%)	967 (33.22%)	3 (0.31%)	50.3% (−146.9 to 89.9)
Consolati et al. [88]	2024	Real World	20 December 2023 to 15 February 2024	Italy, SC (Val d’Aosta)	57	537	369 (68.72%)	0, -	168 (31.28%)	14 (8.33%)	98.4% (72.9 to 99.9)
Moline et al. [61]	2024	Real World (only cases with hospital admission)	1 October 2023 to 29 April 2024	USA, MC	211	699	59 (8.44%)	6 (10.17%)	640 (91.56%)	407 (63.59%)	84.0% (65.8 to 92.5)
Paireau et al. [87]	2024	Real World (only cases with hospital admission)	15 September 2023 to 31 January 2024	France, MC	138	288	58 (20.14%)	37 (12.85%)	230 (79.86%)	201 (87.40%)	73.0% (59.8 to 89.2)
Ares-Gómez et al. [59]	2024	Real World (NCT06180993)	25 September 2023 to 31 March 2024	Spain, MC (Galicia)	154	10,259	9408 (91.70%)	30 (0.32%)	851 (8.30%)	16 (1.88%)	73.0% (69.0 to 90.7)
Ezpeleta et al. [85]	2024	Real World	1 October 2023 to 28 January 2024	Spain, MC (Navarre)	119	1177	1083 (92.01%)	8 (0.74%)	94 (8.99%)	8 (8.51%)	91.3% (77.4 to 96.7)

* Duplicated series.

**Table 3 vaccines-12-00640-t003:** Summary characteristics of studies included in quantitative analysis. Note: LRTIs = lower respiratory tract infections; RR = risk ratio; 95% CI = 95% confidence interval; RCT = randomized controlled trial.

Variable		%	Hospital Admissions Due to LRTIs (n./N., %)	RR	95% CI
Stage	N./45,238				
First year	43,294	95.70%	819, 1.89%	REFERENCE	
Follow-up	1944	4.30%	6, 0.31%	0.163	0.073 to 0.364
Treatment	N./43,249				
Nirsevimab	33,884	78.35%	143, 0.42%	0.058	0.049 to 0.070
Non-treated	9365	21.65%	676, 7.22%	REFERENCE	
Design of the study	N./43,249				
RCT	13,377	30.93%	144, 1.08%	REFERENCE	
Real world	29,872	69.07%	675, 2.26%	2.099	1.756 to 2.510
Length of observation period	N./43,249				
<150 days	18,914	43.73%	216, 1.14%	REFERENCE	
≥150 days	24,335	56.27%	603, 2.48%	2.170	1.860 to 2.532
Status regarding COVID-19 pandemic	N./43,249				
Before COVID-19 pandemic	3797	8.78%	58, 1.53%	REFERENCE	
During or after COVID-19 pandemic	39,452	91.22%	761, 1.93%	1.263	0.969 to 1.646
Settings	N./43,249				
RCT	13,377	30.93%	144, 1.08%	REFERENCE	
Spain	27,112	62.69%	176, 0.65%	0.603	0.484 to 0.751
Other EU countries	2061	4.77%	86, 4.17%	3.876	2.980 to 5.043
USA	699	1.62%	413, 59.08%	54.887	46.132 to 65.303

## Data Availability

Data are available on request to the corresponding author.

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
