# Peer review of "Impact of Nirsevimab Immunization on Pediatric Hospitalization Rates: A Systematic Review and Meta-Analysis (2024)"

_vaccines, 2024, doi:10.3390/vaccines12060640_

Round 1

Reviewer 1 Report

Comments and Suggestions for Authors

This work provides a systematic review of the impact of nirsevimab, a novel antibody, on hospitalization rates for clinical RSV infection based on meta-data analysis. The experimental design was reasonable, the potential bias was considered, and the impact of the results on future research was analyzed in detail. It is helpful to control RSV infection and guide clinical medication. In view of the current attention on the development of RSV vaccine, this article may be of interest to a large readership.

 Two minor suggestions:

1. The title is too long and it is suggested to simplify it;

2. Check the grammar and spelling, there are a few errors. Like line 565, reccomended, should be recommended.

Comments on the Quality of English Language

Good at English.

Author Response

Estimated Reviewer 1,

we warmly thank you for your positive but also collaborative approach. We have accurately amended the main text in accord with your suggestions, providing the revised paper hereby we present.

More precisely:

1. The title is too long and it is suggested to simplify it.

The main title was amended as follows:

Impact of nirsevimab immunization on pediatric hospitalizations: a systematic review and meta-analysis (2024).

2. Check the grammar and spelling, there are a few errors. Like line 565, reccomended, should be recommended.

We apologise for the mistakes we did across the main text. The paper has been revised and errors fixed. Thank you again on the behalf of all Authors.

Reviewer 2 Report

Comments and Suggestions for Authors

Thank you for the opportunity to review this systematic review on the effectiveness of nirsevimab immunoprophylaxis for pediatric RSV infection. The manuscript is thorough and addresses an important topic. However, I have several comments and suggestions for the authors to consider for improving the clarity and robustness of the review.

1. Distinction between efficacy and real-world effectiveness: While the manuscript does an excellent job of summarizing the findings from clinical trials and observational studies, it is crucial to distinguish between efficacy (results observed in controlled clinical trials) and real-world effectiveness (outcomes observed in routine clinical practice). This distinction is important for the readers to understand the potential variability in nirsevimab’s performance outside the controlled environment of clinical trials. Please elaborate on how these differences might impact the interpretation of the results and the potential implications for clinical practice.

2. Stratification of results by exposure to other immunizations: To provide a more nuanced analysis, it would be beneficial to stratify the results based on whether infants were exposed to maternal passive immunization with Abrysvo or whether they received palivizumab. These stratifications could reveal important differences in effectiveness and help identify subgroups that might benefit more from nirsevimab.

3. Clarification of outcomes: RSV-Related LRTI vs. LRTI in general: The manuscript currently does not clearly differentiate between RSV-related lower respiratory tract infections (LRTI) and LRTI in general. Given the focus on RSV, it is critical to define and distinguish these outcomes more explicitly. This will help ensure that the results are interpreted correctly and that the effectiveness of nirsevimab is accurately assessed in the context of RSV-specific outcomes. Please provide clear definitions and ensure that the results are presented separately for RSV-related LRTI and LRTI in general.

Comments on the Quality of English Language

Minor editing of English language is required to improve flow and clarity.

Author Response

Estimated Reviewer 2,

thank you in advance for your accurate and propositive feedback, whose content was not only collaborative but also useful in improving the quality of our study.

We have accurately addressed all of your comments, as follows:

1) Distinction between efficacy and real-world effectiveness.

We agreed with your comments and that the original version of our text was inadequate in addressing this specific topic. We have amended the main text by including the following section:

Efficacy (i.e. the degree to which an intervention prevents a certain outcome) is calculated in RCTs under ideal and controlled circumstances, and even though a certain intervention with documented high efficacy (e.g. COVID-19 mRNA vaccines against original Wuhan strains of SARS-CoV-2) would be expected to be also high performing (i.e. effective) in real-world settings, factors such as how the intervention is delivered, underlying and heterogenous clinical conditions, and actual occurrence of the targeted condition (in this case, the RSV epidemiology) can reduce how effective the intervention was in preventing the disease [89]. According to our data, the actual efficacy of nirsevimab in avoiding hospitalizations due to RSV-associated LRTI was comparable to its real-world effectiveness, stressing how the nirsevimab’s potential impact on the management of RSV disease burden should be taken into consideration not only from a clinical point of view but also from a public health perspective [59,88]. While palivizumab was highly effective in avoiding severe respiratory syndromes associated with RSV infections in highly selected patient groups [50,52,55,90], nirsevimab potentially provides a preventive option to be delivered to the whole of infant population, with a likely impact on the RSV-associated disease burden [57,62,91].

2. Stratification of results by exposure to other immunizations.

Again, we agreed with your observation. In fact, selection criteria of both RCTs and real-world studies mostly and deliberately excluded children exposed to alternative preventive interventions. However, as this specific information was not consistently provided in some studies and most notably the study from Simoes et al. included controls immunized with palivizumab and not receiving a placebo (an issue due to the source of included data), the text deserved some improvements, including a newly designed Table A2. More precisely:

As provided in Appendix Table A2, among retrieved RCTs, treatment of either cases or controls with palivizumab was documented only by Simoes et al. [79], as all controls (n. 290) did actually receive a full conventional immunization course. On the other hand, all sampled children were not exposed to maternal immunization strategy. When dealing with real-world studies, while the studies from Lopez-Lacort et al. [84], Ernst et al. [83], and Ezpeleta et al. [85], did not document about the actual exposure of sampled infants to both palivizumab and maternal immunization, the studies from Consolati et al. [88], Paireau et al. [87], Moline et al. [61], deliberately excluded from their reports all cases managed with strategies alternative to nirsevimab. Finally, the report from NIRSEGAL study [59,86], did not include any case managed with palivizumab, and no information was provided on maternal immunization.

Moreover, in discussion section the following text was included:

For instance, while RCTs likely did not include any patient exposed to maternal immunization strategy, and deliberately excluded cases managed with palivizumab with the notable exception of the MEDLEY subgroup of the study from Simoes et al. [79], reporting strategy from real-world studies is quite more heterogeneous. On the one hand, since the beginning of nirsevimab immunization, delivery of palivizumab has been discontinued in the Spanish region of Galicia [59], but no information is provided from other Spanish regions. On the other hand, the studies of Consolati et al. [88], Moline et al. [61], Paireau et al. [87] deliberately removed from their estimates infants immunized with palivizumab. Similarly, exposure of sampled children to maternal immunization is ruled out for the aforementioned studies, while no information is provided by all of Spanish studies [59,84,85].

...

Because of the original aim of the MEDLEY study, controls did therefore receive a full prophylaxis with palivizumab rather than placebo. Despite the rigorous design of the original RCT, the eventual consistency with other reports mostly focusing on healthy children could be then questioned.

3. Clarification of outcomes: RSV-Related LRTI vs. LRTI in general.

In fact, we warmly thank you particularly for this observation. Our study was aimed to track the occurrence of hospitalizations due to RSV-related LRTI. Double checking our text because of your comment, we noted that some sentences, particularly in materials and methods section, could be misleading. The main text was therefore amended as follows:

introduction:

This study was therefore designed to systematically evaluate the efficacy and the early estimates on effectiveness of nirservimab in infants and children in avoiding hospital admissions due to RSV-associated LRTI, being of possibly guidance for medical and public health professionals.

Methods:

(data extraction 2.5)

(c) Outcome data: definition of primary and secondary outcome; case definition. The main outcome encompassed the reported hospital admissions due to RSV-associated LRTI.

2.7 Data analysis

More precisely, in the present study IE was calculated on the efficacy in avoiding hospital admissions due to RSV-associated LRTI cases

Table 2: the caption "LRTI" was amended to "RSV-associated LRTI cases with hospital admission"

Discussion:

In fact, when provided by the collected study, the proportion of RSV-associated LRTI over the total of hospitalized LRTI (i.e. including both RSV-associated and non-RSV associated cases) ranged between 28.28% (95%CI 19.69 to 38.22) [78] and 100% [85] (Appendix Figure A7), suggesting that the actual proportion of RSV-associated cases may have been somehow underestimated, conversely inflating the pooled estimates for IE. The choice of limiting our analysis on the outcome of hospital admissions due to RSV-associated LRTI likely limited the potential heterogeneity due to the variable case definition of LRTI and diagnostic options from RCTs and real-world settings, being otherwise recommended for the appropriate appraisal of RSV vaccines [48,105] Nonetheless, hospital admission rates for RSV-associated respiratory syndromes have been extensively documented as highly heterogeneous, mostly due to national policies and the availability of healthcare options [19,126–128]. As the largest proportion of included samples from real-world settings were provided by studies from the Spanish settings [115], a cautious approach when generalizing our estimates, not only at global level, but also at the regional (i.e. European) level is therefore required.

Again, we warmly thank you for the whole of your recommendations, and we are confident that the improvements we did perform have improved the overall quality of our paper. 

Round 2

Reviewer 2 Report

Comments and Suggestions for Authors

Thank you for addressing my concerns and improving the manuscript.